# Microstructure and Mechanical Properties of Annealed WC/C PECVD Coatings Deposited Using Hexacarbonyl of W with Different Gases

**DOI:** 10.3390/ma13163576

**Published:** 2020-08-13

**Authors:** Peter Horňák, Daniel Kottfer, Karol Kyzioł, Marianna Trebuňová, Janka Majerníková, Łukasz Kaczmarek, Jozef Trebuňa, Ján Hašuľ, Miroslav Paľo

**Affiliations:** 1Department of Materials Science, Faculty of Industrial Technologies, Alexander Dubček University of Trenčín, I. Krasku 491/30, 020 01 Púchov, Slovakia; hornak.peter@gmail.com; 2Institute of Materials Research, Slovak Academy of Sciences, Watsonova 47, 040 01 Košice, Slovakia; 3Department of Mechanical Technologies and Materials, Faculty of Mechanical Engineering, Technical University of Košice, Mäsiarska 74, 040 01 Košice, Slovakia; janka.majernikova@tuke.sk (J.M.); jozef.trebuna@gmail.com (J.T.); jan.hasul@tuke.sk (J.H.); m.palo@commercservice.sk (M.P.); 4Department of Physical Chemistry and Modelling, Faculty of Materials Science and Ceramics, AGH University of Science and Technology, A. Mickiewicza 30 Av., 30-059 Kraków, Poland; kyziol@agh.edu.pl; 5Department of Biomedical Engineering and Measurement, Faculty of Mechanical Engineering, Technical University of Košice, Letná 9, 042 00 Košice, Slovakia; marianna.trebunova@tuke.sk; 6Institute of Materials Science and Engineering, Lodz University of Technology, 1/15 Stefanowskiego Str., 90-924 Łódź, Poland; lukasz.kaczmarek@p.lodz.pl

**Keywords:** coating, PECVD, tungsten carbonyl, annealing, mechanical properties, tribological properties

## Abstract

The present work studies the tungsten carbide (WC/C) coatings deposited by using Plasma Enhanced Chemical Vapor Deposition (PECVD), with and without gases of Ar and N_2_. Volatile hexacarbonyl of W was used as a precursor. Their mechanical and tribological properties were evaluated. The following values were obtained by using deposition process with N_2_ of H_IT_ = 19.7 ± 4.1 GPa, E_IT_ = 221 ± 2.1 GPa, and coefficient of friction (COF) = 0.35 ± 0.09. Secondly, deposition without the aforementioned gas obtained values of H_IT_ = 20.9 ± 2 GPa, E_IT_ = 292 ± 20 GPa, and COF = 0.69 ± 0.05. WC/C coatings were annealed at temperatures of 200, 500, and 800 °C, respectively. Evaluated factors include the introduced properties, the observed morphology, and the structural composition of WC/C coatings. The process of degradation was carried out by using various velocities, depending on used gases and annealing temperatures.

## 1. Introduction

Tungsten carbide is oftentimes used in the thin coating form and is known as a solution with high hardness and durability against abrasive wear. Due to the already mentioned properties, WC/C coatings have a low coefficient of friction (COF) and are used as a protection of functional surfaces of mechanical components made out of steels and for coating of high-speed steels, as well as hard metals. WC coatings are deposited by using PVD (Physical Vapor Deposition) and CVD (Chemical Vapor Deposition) methods concerning the high temperature of fusion of WC (*ca.* 2550 °C) [1]. Out of all PVD methods, the most used methods are direct current magnetron sputtering (DCMS) [2,3,4,5,6,7,8,9,10] and radiofrequency magnetron sputtering (RFMS) [9,11,12,13,14,15] with Ar. Reactive sputtering of W, C, or WC from a target, in the presence of hydrocarbon [2] or other added gases [3,4,14], is used, too. Other effective methods of sputtering include High Target Utilization Sputtering (HiTUS) [15] and High-Power Impulse Magnetron Sputtering (HiPIMS) [3]. WC coatings can be deposited by using the CVD method [16,17]. A disadvantage of the CVD method is its high depositing temperatures, ranging from 800 to 1200 °C, which exclude, for example, aluminum alloy from being coated. It is often necessary, following the process of coating of steels, to provide thermal treatment of said coatings. Gesheva et al. [18], using pyrolytic decomposition of W(CO)_6_, at 400 °C, in argon atmosphere, deposited WC thin films on Si substrate. Gesheva et al. also deposited polycrystalline W films [18], using hydrogen reduction of WCl at 750 °C, in Ar atmosphere. The deposition of coatings by the decomposition of W and Mo carbonyls, using the CVD technique, has been published in References [18,19], and the same process by the decomposition of Cr carbonyl in Reference [20]. Gesheva et al. [18] obtained thicknesses of the WC coatings ranging from 0.12 to 0.6 µm and MoC coatings from 0.12 to 0.3 µm. The temperature of deposition was from 340 to 400 °C. Electric properties were evaluated. Sagalovich et al. [19] deposited MoC coatings in the temperatures from 480 to 540 °C and pressures in the vacuum chamber ranging from 9 up to 16 Pa. The hardness of the obtained MoC coating was situated in an interval from 11 to 18 GPa, while the thickness was presented by using the cross-sectional scanning electron microscopy (SEM) view of the pictures, from 143 to 173 µm. A CrC coating was deposited by Erokhin et al. [20], using the process of decomposition of Cr carbonyl. Hardness was equal to 17 GPa; meanwhile, the thickness of the coating was not stated. The PECVD method involves using Ar to create low-pressure and low-temperature plasma, which enables the deposition of a coating in temperatures up to 500 °C, where W hexacarbonyl [21,22,23,24] has been used as a precursor. Usoltsev et al. [25] explored the process of creation and decomposition of W(CO)_6_ and Mo(CO)_6_. Rezuchina and Švyrev [26] and Garner et al. [27] researched the pressure of saturated gases and their sublimation temperatures. The dependence of pressure of W(CO)_6_ gases on temperature, according to Garner et al. [27] and Chelappa and Chandra [28], is given by the following equation: log *p* = 10.65 − (3872/T), where *p* is pressure in kPa, and *T* is the absolute temperature in K. Pressure of W(CO)_6_ gases, in temperatures ranging from 20 to 40 °C, ranges from 2.7 to 19 Pa. This comes as a result of the equation above. This is also allowing us to achieve relatively high velocities of sublimation while working pressures in the chamber of the PVD apparatus are ranging from 0.01 to 10 Pa. In the aforementioned, it is also implied that the usage of carrier gas is not necessary. The temperature of the decomposition of W(CO)_6_ is 170 °C [21]. Dependence of coefficient of friction (COF) and H_IT_ WC/C coatings on the content of C in the coating and dependence of H_IT_ and COF on the volume of *a*-C in the WC/C coating have been identified by El Mrabet et al. [9]. Gases such as H_2_ [2,22] and C_2_H_2_ [22] are often used to affect the mechanical and tribological properties of WC/C coatings. Other effective gases and mixtures of gases are not used very often in the process of deposition of WC/C coatings using PVD and CVD methods, since there is a lack of information about such description phenomena in the literature. The usage of W carbonyl for deposition of WC/C PECVD coatings and evaluation of their properties (H, E, and COF) is not published fairly often, too. That is the reason why W(CO)_6_ has been used as a precursor and added gases Ar or N_2_ in the process of deposition of WC/C coating, using the PECVD method. This work aimed to evaluate the effect of temperature on the thermal stability, hardness, Young′s modulus, and COF of the WC/C coatings, with and without the added gases, such as Ar and N_2_. Obtained results were compared with other published results and results from articles published by us [4,14], to which this work relates.

## 2. Materials and Methods

### 2.1. Sample Preparation

Two types of samples were prepared. Samples of a monocrystal of Si with approximate size 20 × 15 × 1 mm were used to evaluate the thickness and structure of WC/C coatings via scanning electron microscopy (SEM) JEOL JSM 7000F Tokyo, Japan. The sample can be easily broken post-deposition. The breakage must be carried out in a manner wherein the coating is exerted by using pulling forces. Afterward, the breakage can be inspected by using SEM. Samples for tribological tests were made out of steel C45 (STN 412050). The chemical composition (wt.%) of steel samples was per Technical Standards (STN EN 412050): 0.42–0.50% C, max 0.40% Si, 0.50–0.80% Mn, max 0.40% Cr, max 0.10% Mo, max 0.40% Ni, max 0.035% *p*, and max 0.035% [4]. Samples were made by using wire electrical discharge machining (WEDM) from bars of circular cross-section profiles with diameters of 50 and 25 mm. Function surfaces were machined to a thickness of 3.00 ± 0.05 mm. The substrate was tempered in oil to the temperature of 860 °C, followed by annealing at the temperature of 200 °C. Following heat treatment, the substrates were polished in a diamond paste with the granularity of 15, 9, and 3 µm. Subsequently, the substrates were polished specularly by 1 µm of diamond paste, to the final roughness of the surface being R_a_
*ca.* 12 nm. The substrates were then purified with ultrasound, in an acetone environment for 10 min, and dried with an electric hairdryer for 5 min. After that, the substrates were inserted into the vacuum chamber. Before deposition of the coating and after draining the vacuum chamber, the substrates were cleaned by etching in argon, in glow discharge. The parameters of the cleaning process were as follows [4]: pressure (2 Pa), bias of the holder U_b_
*ca.* −5 kV; and current density to the substrate samples (1 mA∙cm^−2^) and the time of cleaning (15 min). In addition, Ar with a flow of 65 cm^3^∙min^−1^ was blown into the vacuum chamber.

### 2.2. Deposition of Coatings and Annealing

A ZIP 12 device was used for deposition of the WC/C coating with sublimation chamber (Scheme 1). The PECVD method was used with direct current. Moreover, the negative electric potential was connected to electrically conductible substrates against the metallic vacuum chamber of the PVD apparatus. An auxiliary anode was located in the chamber. The substrates were placed onto the substrate holder; the crucible and the precursor were placed into the sublimation chamber and enclosed by a valve between sublimation chamber and vacuum chamber. After the cleaning procedure and ensuring the required yield pressure, which was *ca.* 10^−3^ Pa, the deposition of WC/C coating was carried out. Relating to the experiment presented in this paper, argon (Ar) with a purity of 99.999% was used as a carrier gas.

The valve between the sublimation and vacuum chamber was opened. Pulverized volatile W(CO)_6_ (W hexacarbonyl) was used as a precursor for the PECVD process of deposition of the WC/C coating. During the process, the sublimation of vapors of W(CO)_6_ (it is based on the chamber vacuum and temperature of precursor [27,28]) into the surroundings between substrate holder with substrates and auxiliary anode, where these vapors ionize (ionized gas is also called low-pressure, low-temperature plasma). Ionized particles of the sublimed precursor were accelerated at a substrate passing through the plasma.

Decomposition of W(CO)_6_ to W + 6 CO and 2 CO → C + CO_2_ reaction takes place in the plasma area and on the surface of the substrate. Carbon created in this manner reacts with tungsten and creates tungsten carbide. The real mechanism of the decomposition created by the collision of W(CO)_6_ molecules with electrons and their ions is significantly more difficult and the existence of W(CO)_6_ fragments of the decomposition in the excited and ionized state can be assumed.

For control of the deposition temperature, the substrates were covered with Kapton tape before the deposition of the WC/C coating. Kapton tape is made out of polyimide with a silicone sticking surface on one side. It is characterized by heat-resistance, depending on the manufacturer, up to the temperature of 350 °C.

Annealing of samples with WC/C coating was carried out by inserting the substrates into an electric oven, at temperatures of 200, 500, and 800 °C, without the protective atmosphere, for 1 h. After that, the oven was turned off, and the substrates cooled down naturally. Annealing at a temperature of 200 °C was carried out only to assure the effect of inner tension on the indentation hardness.

### 2.3. Nanohardness and Young′s Modulus

H_IT_ hardness and indentation module E_IT_ on the coated substrates were explored by using instrumented indentation on nano hardness tester (NHT), CSM Instruments, Basel, Switzerland on steel substrates. The conditions of measurements were as follows: sinus mode with 1 mN amplitude, encumbered strength was established according to the thickness of the coating in an interval from 20 to 60 mN, frequency 15 Hz. A diamond Berkovich indenter was used. Ten measurements were carried out on all of the WC/C coatings. Curves with an extreme course were excluded from the analyses. Values of indentation hardness and indentation module were calculated as an average of maximal values of indentation curves.

### 2.4. SEM, AFM, XRD, and GDOES Analyses

Thickness, the morphology of the surface of the evaluated coatings, and their microstructure were observed by an electron microscope JEOL JSM 7000 F, Tokyo, Japan.

Chemical analysis was evaluated as the depth of the chemical concentration profile, using the Glow Discharge Optical Emission Spectroscopy (GDOES) technique (Leco Corporation, St. Joseph, MI, USA). During this analysis, the Spectruma GDA 750 equipment was applied, which is provided with a monochromator with a Czerny–Turner design (focal length of 480 mm). The spatial resolution of the monochromator was 0.025 nm.

The phase composition of WC/C coatings was measured by using diffraction analysis. The spectrums were gauged by a diffraction meter X′Pert PRO Philips, Eindhoven, Netherland with a detector X′Celerator in Bragg–Brentanto parafocusational arrangement with a regime of measurement θ/2θ. The source of RTG (X-ray diffraction) radiation was a copper cathode (I = 40 kV, U = 50 mA) with characteristic RTG radiation CuKα_1,2_ in a wavelength 1.5418 × 10^−10^ m. XRD spectra were measured in a range of angles 10–100°, with a step length of 0.033°. Qualitative analysis was carried out with the help of the CMPR program, a power diffraction toolkit [29], and interpreted with the PDF-2 database [29].

The topography of the coated surface was measured by using an atomic force microscope (AFM), Dimension Icon, Veeco, Plainview, NY, USA.

### 2.5. Coefficient of Friction

The COF of WC/C coatings prepared on steel substrates was measured by a Ball-on-Disc method, using a high temperature tribometer (HTT), CSM Instruments, Needham, MA, USA, in a load of 0.5 N and room air temperature of 21 °C. A steel ball 100Cr6 with a diameter of 6 mm, a velocity of 10 cm/s, and a sliding distance of 50 m was used as a counterpart. The coefficient of friction and the depth of penetration of the substrate surface were continually noted down for each test as a function of time, a number of turnings, and distance. Wear of evaluated coatings and wear of the counterpart (ball) were not evaluated.

## 3. Results and Discussion

The WC/C coatings were deposited in the following combinations: (i) optimization of the deposition parameters concerning the maximal hardness and minimal COF without the use of added gas (only using sublimed carbonyl of W), (ii) deposition of the optimized WC/C coating with added gas of Ar, and (iii) deposition of the optimized WC/C coating with added gas of N_2_. Each WC/C coating that was deposited with added gases was evaluated after deposition and after annealing at the temperatures of 200, 500, and 800 °C, respectively.

### 3.1. Optimization of the Deposited Coatings′ Parameters

The results of the optimization of the deposition parameters considering maximal hardness and minimal COF value, without the usage of added gas, while only using sublimed gas W carbonyl, are below in Table 1.

Table 1 shows that H_IT_ values of WC/C coating deposited without Ar (carrier gas) were in an interval from 14.2 to 20.9 GPa. The highest measurement of hardness occurred at a pressure of 3.0 Pa. E_IT_ values were from the interval of 180–305 GPa.

The lowest value of Young′s modulus was measured, while the pressure in the vacuum chamber was at 2.0 Pa, and the highest was measured at 3.0 Pa. Since bias voltage and current density were constant, it can be stated that pressure in the vacuum chamber has a detrimental effect on evaluated hardness and Young′s modulus. This also applies to COF. The maximum value of COF was equal to 0.8 with a pressure of 1.0 Pa, and the minimum value was equal to 0.23 with the pressure of 3.0 Pa. The deposition parameters of the WC/C coating with the maximum value of H_IT_ were accepted for the next experiment.

Moreover, WC/C coatings were prepared with a carrier gas of Ar, or with an addition of N_2_. The total pressure of deposition was 3 Pa when no added gases were added (using only sublimated carbonyl). For deposition with added gases in the chamber (Ar or N_2_), the pressure was 2 or 4 Pa. The ratio of partial pressures of added gases and the precursor was 1:1. The same amount of precursor was used for the preparation of coatings—5 g. The time duration of the deposition was 2 h. Table 2 summarizes the specifications of the deposition and mechanical properties of the prepared coatings. The mentioned properties were evaluated before annealing.

#### 3.1.1. Microstructure

WC/C coatings evaluated by SEM analysis, prepared by using the PECVD method (Figure 1), on steel substrates without added gas, have a so-called cauliflower structure, as in Reference [23], with the size of cauliflower globulites ranging from 500 nm to 2.0 µm. The thickness of WC/C coatings was ca. The topography of coatings shown by using AFM confirmed the globulitic particles (Figure 2). Their creation is attributed to the circumstances of nucleation and the growth of coatings. The explanation for this kind of topography is the presence of grain structure in the lower part of grains, which are finished with the branching of globulite grains with the so-called cauliflower structure. These formations can be made out of substructures, such as nanocolumns or nanocauliflowers [23].

#### 3.1.2. Indentation Hardness and Young′s Modulus

Figure 3 shows depth profiles of the indentation hardness course of examined WC/C coatings deposited with and without Ar (with maximum values of indentation hardness). All of the profiles show approximately the same course: The growth of the curves begins in extremely small depths with the hardness of 10 GPa. This growth reaches a maximum in depths of penetration *ca.* 50–60 nm, where the coatings′ thickness was *ca.* 0.7 µm. After that, the curves have a descending character. The peak of these curves should not be in the depth deeper than 1/10 of the coating thickness. Therefore, the resulting value of indentation hardness will not be affected by the hardness of the substrate. The WC/C coating deposited with additive gas of Ar (obtained by optimization) has the highest indentation hardness, at H_IT_ = 28.5 ± 2 GPa, and Young′s modulus, at E_IT_ = 351 ± 28 GPa. The hardness of a WC/C coating deposited with N_2_ is equal to H_IT_ = 20.9 ± 2 GPa and Young′s modulus E_IT_ = 298 ± 20 GPa. WC/C coatings prepared with argon in the PECVD process are reported to have the highest indentation hardness, *ca.* 28 GPa, which is correspondent to the hardness of WC/C coating with the content of C equal to *ca.* 33% [9]. Furthermore, the presence of C in the coating was not evaluated. It can be assumed, based on constant technological parameters of the WC/C coatings′ deposition (time duration of the deposition was equal to 120 min), that the thickness of the coating was close to the thickness of the coating deposited with added gases Ar and N_2_ (see Section 3.2.1 and Section 3.3.1, respectively). Moreover, it can be expected that the thickness of the WC/C coating (deposited without Ar) is close to these values. The mentioned supports the development of hardness (Figure 3), where the maximum measured hardness is in depths of ca. 50 nm.

#### 3.1.3. Coefficient of Friction

The COF of the optimized WC/C coating reached the value of 0.26 (Figure 4), which corresponds with the COF of the WC/C coating with *ca.* 58% of C [9]. However, the content of C in the coating was not evaluated. Figure 4 exhibits the course of WC/C coating with COF = 0.8 which was deposited at the argon pressure of 1.0 Pa. The coating’s high value of hardness (*ca.* 28 GPa) is accompanied with a high value of COF, which can be attributed to the presence of WC and a lower content of graphite.

### 3.2. Ar Effect

#### 3.2.1. Morphology, Phase Analysis, and Chemical Composition

The surface reports a nano-column structure and presence of cauliflower globulites (see arrows) on the surface with size up to 100 nm (Figure 5a), while the cross-sectional view is pointing toward a dense columnar structure (Figure 5d). On the surface, following annealing with a temperature of 500 °C, the globulite particles situated on the surface slightly thickened. However, the character of the surface remained the same. On top of that, clusters of particles with a size of ≈200 nm (Figure 5b—arrows) started to form locally. After annealing with the temperature of 800 °C (Figure 5c), a visible disintegration of the structure occurred. Large spherical particles of WO_3_ with a diameter upward of 0.5 µm can be spotted on the surface. An effect that is similar to the one on WC/C coating by depositing Ar [4] manifested. The dark places represent the surface without WC/C coating. These empty spots usually have size up to 50 nm (at some places, up to 100 nm) and are recognizable even at a lower resolution. Empty spots are a consequence of WC/C coating disintegration. This disintegration is affected by the presence of chemical elements in the air. There is a rupture in the structure as a result of the swelling process—relating to the second phase, where O_2_ (penetrating the coating) reacts with C and W [30] to produce CO_2_ and WO_3_, respectively. The coating deposited with Ar is 0.7 µm thick (Figure 5d).

The chemical composition of the deposited coating (Figure 6) shows that the amount of W ca. 95% is constant on the surface of the coating down to the depth of ca. 0.45 µm. The content of C in the coating is linearly decreasing from *ca.* 44% on the surface down to ca. 20%.

Phase analysis of WC/C coating (Figure 7) in a non-annealed state (room temperature-RT) shows the presence of carbide phase WC_1−x_ and WO_3_. After annealing at the temperature of 500 °C, the structure undergoes amorphization. Only a subtle change of the intensity of the peaks occurs—a decrease of amorphous WC_1−x_, as well as a WO_3_ peak. By annealing at the temperature of 800 °C a significant reaction of the coating and the environment in the annealing furnace takes place. A sharp carbide phase of WC_1−x_ and WO_3_ is present (respectively), as well as a bold phase of W_2_N [30]. Given the fact that WC coatings crystallize at higher temperatures, a bold W_2_N phase could appear in the structure, according to Choy K.L. [31].

#### 3.2.2. Mechanical and Tribological Behavior

H_IT_ and COF of WC/C coatings, depending on annealing temperature, with and without doped gases, are presented in Figure 8. Ar as added gas caused the growth of hardness, as opposed to the coating deposited without the added gas. Hardness (RT) grew to *ca.* 28 GPa (Figure 8a), which corresponds to the H_IT_ of the WC/C coating with C *ca.* 34% [9], which also is a tabulated hardness of WC *ca.* 28 GPa [1]. However, the evaluated coating contains *ca.* 44% C (Figure 6). According to El Mrabet et al. [9], a WC/C coating with this amount of C can have hardness equal to 18 or 32 GPa. The difference can be attributed to the presence of a-C in the coating and presence of carbide phase WC_1−x_ (Figure 7). The achieved hardness (*ca*. 28 GPa) is 33% lower when compared to Horňák et al. [4] (37.2 ± 4.8 GPa), who used the DCMS method with the carrying gas of Ar, and 22% higher when compared to Horňák et al. [14] (22.2 ± 1.3 GPa), who used the RFMS method with the carrying gas of Ar.

Ar was implemented during the deposition into the WC/C coating, which could have caused a significant growth of hardness as a result of the generated tensions to the crystal lattice. Nevertheless, the research on the amount of Ar in the coating has not been evaluated. The influence of Ar in the WC/C coating on the hardness and COF could be a topic for future research.

Hardness slightly decreased to 26.0 ± 1.5 GPa after annealing at a temperature of 200 °C, which is equal to the minimum measured value before annealing. On the other hand, after annealing at the temperatures of 500 and 800 °C, the hardness of the WC/C coating has significantly decreased under 5 GPa (3.0 and 2.5 GPa—Figure 8a). These values are 50% less than the values measured by Reference [4], yet in good agreement with Reference [14]. The hardness of WC/C coating after annealing at the temperature of 800 °C fits the status of its important disruption as a result of crystallization of the original amorphous structure, which fits Reference [5].

The COF value of the deposited WC/C coating with Ar grew to 0.77 ± 0.02 (Figure 4), which is three times more when compared with the coating deposited without Ar. It also is 26% more than what Horňák et al. reached, using DCMS [4] (0.64 ± 0.09) and close RFMS (*ca.* 0.82) methods. The value of COF, which we measured, is close to the value given by El Mrabet, in Reference [9].

The high value of COF is a consequence of the value of H_IT_ = 28 GPa, which is also considered to be relatively high, when compared to the hardness of the counterpart (a sphere made out of steel 100Cr6) used in the tribotest. After annealing at the temperature of 200 °C, COF has decreased to 0.36 ± 0.06 and after annealing at the temperature of 500 °C, a tremendous decrease of the value of COF occurred, down to 0.27 ± 0.05. That is one-third more than in Reference [14], and 25% less when compared to Reference [4]. Ar inhibited the degradation of WC coating by swelling (Figure 5b). That could have caused the creation of a smaller contact surface between WC coating and the counterpart (ball) during the Ball-on-Disc test, which could have reduced the value of COF when compared with WC coating with N_2_ (Figure 8a).

COF increased to the value of 0.36 ± 0.05 after annealing at a temperature of 800 °C, which is the same value as annealed at a temperature of 200 °C (Figure 8b). A significant decrease of the COF after annealing at the temperatures of 500 °C (see Figure 8c, using N_2_) and 800 °C (see Figure 8d, using Ar) could have been caused by the presence of C in the coating in the form of graphite. Another reason is a strong decrease in hardness of the evaluated coating after each process of annealing. Tribological properties of magnetron sputtered WC/C coatings, concerning the amount of C in the coating, were evaluated by Mrabet et al. [9]. However, the determination of the amount of C in the form of graphite is necessary. The mentioned criterion is often important when evaluating the sources of the increase/decrease of H_IT_ and COF of the WC/C coatings obtained when using the PVD and/or CVD methods.

### 3.3. N_2_ Effect

#### 3.3.1. Morphology

As per Figure 9a, the surface of the WC/C coating is smooth, with column-like nanoparticles, with a diameter under 20 nm. The cross-sectional view has confirmed this visible structure (Figure 9b). The thickness of the coating is *ca.* 0.5 µm, while annealing at a temperature of 500 °C caused a change in the morphology. Clusters of oxides formed, with the diameter ranging from 300 nm to 1 µm (Figure 9c). Moreover, the thickness of the layer reached *ca.* 0.7 µm (Figure 9d). After annealing at the temperature of 800 °C, oxidic clusters reached a diameter of 2.0 µm and more (Figure 9e). Oxidation formed on the surface of the WC/C coating, with a thickness of 100 nm, considering the total thickness of the layer being *ca.* 0.7 µm (Figure 9f).

Chemical composition of the coating (Figure 10) shows that the content of W is slightly decreasing from the surface of the coating, from the value of *ca.* 95% down to *ca.* 91%. The presence of C in the coating decreases linearly from the value of *ca.* 40% on the coating’s surface, to *ca.* 20%. The amount of N_2_ on the surface decreased *ca.* 5% down to the value of *ca.* 1% on the substrate-coating line. O_2_ is present only on the surface of the coating, which could be caused by oxidation of the coating in the air.

#### 3.3.2. Phase Analysis

Before annealing, the WC/C coating showed its amorphous character with nanostructured morphology (Figure 11). Phase analysis of the coating after annealing at a temperature of 500 °C is referring to its crystal character. The carbide phase of WC_1−x_ and WO_3_ and a wide amorphous peak are present in the coating. A similar character is shown by an RTG recording after annealing at a temperature of 800 °C. Maximums of amorphous and carbide WC_1−x_ and oxidic WO_3_ decreased. A glassy amorphous structure can be shown by using a cross-sectional view, which is in accordance with Reference [5], with a suggestion of the creation of crystal structure (Figure 9f).

#### 3.3.3. Mechanical Properties and COF

The measured hardness H_IT_ = 28 GPa was in agreement with the case of the WC/C coating deposited with Ar (Figure 8a), which is one-fifth less as in Reference [4] (24.5 ± 1.2 GPa), where DCMS with N_2_ added gas was used, and 50% less when compared to Reference [14] (13.5 ± 1.0 GPa), where RFMS with N_2_ carrying gas was used. Hardness decreased by 45%, down to 10 GPa, after annealing at a temperature of 500 °C, which is much more than in the argon process. After further annealing at temperatures of 500 and 800 °C (respectively), hardness slightly increased to 12 GPa. N_2_ in deposited WC/C coating caused a decrease of COF of the evaluated coating deposited with Ar, from 0.77 ± 0.02 down to 0.59 ± 0.03. That is 45% less, compared to the coating deposited with Ar (Figure 8b). It is simultaneously *ca.* 50% less when compared with what was measured in Reference [4] (0.26 ± 1.2 GPa) and Reference [15] (0.23 ± 0.2 GPa). On the other side, COF deposited with the W carbonyl without added gas is 50% lower. After the process of annealing at temperatures of 200, 500, and 800 °C, respectively, COF decreased to 0.37 ± 0.06, which is close to the WC/C coating deposited with the Ar. A decrease in COF could be caused by a decrease in hardness after annealing, while a decrease in hardness after annealing could have caused an increase in the content of C in its graphite form, which acts as a dry lubricant.

## 4. Conclusions

Based on the obtained results, the following could be said:Maximal values (deposition with W carbonyl as precursor) were H_IT_ = 20.9 ± 2 GPa and E_IT_ = 298 ± 20 GPa, with a bias voltage of −5 kV and pressure of 3.0 Pa. The COF was 0.3.Additional gases, such as Ar and N_2_, significantly increased values of H_IT_. COF value in the case of additional Ar increased more than twofold. Added N_2_ caused a marked increase of COF at a slight increase in the value of hardness H_IT_.Sample No. 6 with WC/C coating deposited with added gas Ar is the best sample according to values of hardness and COF. Its hardness is equal to 28.5 ± 2 GPa, which is 33% greater, as opposed to the coating deposited without added gas (*ca.* 20.9 GPa). On the other hand, the COF of the coating deposited with Ar (*ca.* 0.8) is more than three times greater when compared with the coating deposited without added gas (*ca.* 0.26). Therefore, when choosing one of these coatings, it is necessary to consider the conditions of the components′ coated surface usage.During deposition, Ar was built into the coating, which could have caused internal compressive pressure. That could have, in turn, caused an increase of hardness, the opposite of the coating deposited without added gas. In the case of N_2_, H_IT_ may increase due to the formation of W nitrides in the coating, which has a higher H_IT_ than W.The temperature of annealing in the case of added N_2_ and Ar caused a dramatic decrease of H_IT_. In the case of Ar, the measured H_IT_ values were significantly smaller.In the case of added Ar, the 500 and 800 °C temperatures of annealing caused a dramatic decrease of COF, from 0.77 ± 0.02 to 0.27 ± 0.05 and 0.37 ± 0.06. In the case of added N_2_, the annealing temperatures caused a similar decrease in COF.In the process of annealing in an unprotected atmosphere at the temperatures of 500 and 800 °C, the WC/C coatings were degraded mainly due to oxidation, which was partially accompanied by swelling. WC/C coating deposited with added Ar and annealed at the temperature of 800 °C was significantly degraded due to the aforementioned mechanisms.It is appropriate to use N_2_ as an added gas, to improve the resistance against oxidation of the WC/C coating deposited using the PECVD method with W hexacarbonyl.

In the future, it could be useful to aim at the research of the influence of Si in the form of added gas SiH_4_ on the mechanical and tribological properties of the WC/C coating. Another interesting topic of research could be the influence of built-in Ar in the WC/C coating deposited from the carbonyl of W on the mentioned properties.

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
