# Peer review of "Microstructure and Mechanical Properties of Annealed WC/C PECVD Coatings Deposited Using Hexacarbonyl of W with Different Gases"

_materials, 2020, doi:10.3390/ma13163576_

Round 1

Reviewer 1 Report

This manuscript has discussed about the preparation of PECVD coatings of WC/C and studied its physical properties. It is written nicely and would be useful for industrial applications. It can be accepted for publication with minor comments, some of which are given below.

  1. Line 104-105: it seems repetition of line ‘after draining the 104 vacuum chamber’
  2. The WC/C coatings without carrier gas: how the volatile species move into deposition chamber without carrier gas?. Is it based on the chamber vacuum condition or by any other mechanism?.
  3. In table 1, there were commas used instead of decimal points (ex. 16,7) in HIT and EIT values in the table and text. Also, EIT values were in the range of 180-305 (not 307, Line 187).
  4. The authors may refer the valules of HIT, EIT and COF values of WC/C coatings prepared with sputter coatings (made of WC sputter target) form their or literature works in order to compare the resultant coatings parameter.
  5. Among the samples, which is better based on the hardness and COF?.
  6. What is the thickness values with and without Ar gas deposition?.
  7. What could be the reason for the increase of HIT due to additional gases, any interaction mechanism possible?.

Author Response

Response to Reviewer 1 Comments

Thank you very much for your kind evaluation of our work. We do agree with all your proposals and comments and have modified the manuscript according to them.

We hope that with your comments the manuscript will be suitable for publishing in Materials and will attract many potential readers of the journal.

Kind regards,

Authors.

Comments and Suggestions for Authors:

This manuscript has discussed about the preparation of PECVD coatings of WC/C and studied its physical properties. It is written nicely and would be useful for industrial applications. It can be accepted for publication with minor comments, some of which are given below.

  1. Line 104-105: it seems repetition of line ‘after draining the 104 vacuum chamber’

Response: Thank you. Line 108 - is revised (in blue)

  1. The WC/C coatings without carrier gas: how the volatile species move into deposition chamber without carrier gas?. Is it based on the chamber vacuum condition or by any other mechanism?.

Response: Thank you. It is based on the chamber vacuum and temperature of the precursor. Carbonyl of W sublimes at room temperature [28,29]. Velocity of sublimation is higher, when pressure is equal to 10-3 Pa. The process of lowering the pressure to the required level and the etching process have taken 40 minutes. The required amount of gas, which has been allowed to enter the vacuum chamber by opening the valve between the sublimation chamber and vacuum chamber, has sublimed during the mentioned processes. The sublimation chamber is located on the door of the vacuum chamber; the connection valve is directly located next to the auxiliary anode. Besides, the carrier gas is not needed for the transportation of the gas of carbonyl of , which is ionized and accelerated in the direction of the coated surface of the sample. Lines 126, 127, 130 - are revised (in blue).

  1. In table 1, there were commas used instead of decimal points (ex. 16,7) in HIT and EIT values in the table and text. Also, EIT values were in the range of 180-305 (not 307, Line 187).

Response: Thank you. Table 1 and Line 192 - are revised (in blue)

  1. The authors may refer the valules of HIT, EIT and COF values of WC/C coatings prepared with sputter coatings (made of WC sputter target) form their or literature works in order to compare the resultant coatings parameter.

Response: Thank you. Yes, although the suggested comparison could be intricate due to the amount of references in Mrabet et al. [9]. That is why the measured features (in sections 3.2.2. and 3.3.3.) were compared to our works [4,15], of which this work is a continuation. We have also compared the measured results with Mrabet et al. [9], which is about the dependency of HIT and COF on the percentage of C in the WC coatings. In [9], the dependency of HIT and COF on the percentage of C is presented in graphs as a summary of the published articles. It seems more comprehensible to us in this manner. It can also be said that the PECVD deposition process does not use a target because it is not a sputtering. On the other hand, the authors of articles on sputtered coatings often do not give the exact chemical composition of the target (in at % and wt.%). Also, lines 302, 303, 318, 369-372, 377 and 378 - are revised (in blue).

  1. Among the samples, which is better based on the hardness and COF?.

Response: Thank you. According to hardness and COF, the best sample is sample number 6 (with the WC/C coating, deposited with added gas of Ar). Its‘ hardness is 28.5±2, which is 33% more than the coating deposited with the added gas of Ar (ca. 20.9 GPa). On the other hand, COF of the coating deposited with Ar (ca. 0.8) is more than three times higher as opposed to the coating deposited without the added gas (ca. 0.26). Due to that, during selection of one of those coatings, it is neccesary to acknowledge the circumstances of the usage of the deposited surface of the component and hardness of the material out of which the counterpart has been made - Lines 392-397 – are supplement, revised (in blue)

  1. What is the thickness values with and without Ar gas deposition?.

Response: Thank you. Thickness of WC/C coating deposited without Ar was not measured. Based on the constant technological parametres of the WC/C coatings‘ deposition (duration of deposition: 120 minutes), we can assume that the thickness of the coating was close to the thicknesses of the coating deposited with added gases of Ar and N2. The coating deposited with Ar is 0.7 µm thick (Figure 5d) and the coating deposited with N2 has thickness equal to 0.5 µm (Figure 9b). Following this, we can assume that the thickness of the WC/C coating is close to these values. The mentioned also supports the development of hardness (Figure 3), where the maximal measured hardness is in depth of ca. 50 nm. Lines 231, 240-246 – are supplement, revised (in blue)

  1. What could be the reason for the increase of HIT due to additional gases, any interaction mechanism possible?.

Response: Thank you. During depositon, Ar has been built into the coating, which could have caused internal compressive voltage. That could have, in turn, caused an increase of hardness, opposite of the coating deposited without added gas. In the case of N2, HIT may increase due to the formation of W nitrides in the coating, which have a higher HIT than W. Lines 398-401 – are revised (in blue).

Reviewer 2 Report

The manuscript studies the effect of coating steels. It could be interesting but there are some issues to be addressed.

ABSTRACT: avoid the use of acronym, since someone has to read the text to understand the values of the abstract

MATERIALS AND METHODS:

  • Which type of indentator? materials? (as it is specified for the tribomechanical tests).
  • How to prepare the samples for SEM observation must by specified. It is necessary and can affect the results, mainly under the cross-section view
  • It is said confocal: where are these results?
  • They prepared to type of samples: over Si and over steel, when are used each type?

RESULTS AND DISCUSSION:

  • Line 213-215: it is said columns. However, they are not clear in such image. Neither AFM image provides data to assure such structure.
  • Figure 9: the figure caption is not clear. X-ray diffraction scan? What is it?
  • How has been the chemical analysis performed?

Author Response

Response to Reviewer 2 Comments

Thank you very much for your kind evaluation of our work. We do agree with all your proposals and comments and have modified the manuscript according to them.

We hope that with your comments the manuscript will be suitable for publishing in Materials and will attract many potential readers of the journal.

Kind regards,

Authors.

Comments and Suggestions for Authors:

The manuscript studies the effect of coating steels. It could be interesting but there are some issues to be addressed.

ABSTRACT: avoid the use of acronym, since someone has to read the text to understand the values of the abstract

Response: Thank you. ABSTRACT - Line 24 - is revised (in red).

MATERIALS AND METHODS:

  • Which type of indentator? materials? (as it is specified for the tribomechanical tests).

Response: Thank you. A diamond Berkovich indenter was used. Line 150 - is supplement and revised (in red).

  • How to prepare the samples for SEM observation must by specified. It is necessary and can affect the results, mainly under the cross-section view

Response: Thank you. Samples of monocrystal of Si have been used to evaluate the thickness and structure of WC/C coatings via scan electron microscopy (SEM). The sample can easily be broken after deposition. The breakage must be carried out in a way, in which the coating is stressed in a pulling motion. After that, the breakage can be observed using SEM. Lines 92 -  95 - are supplement, revised (in red).

  • It is said confocal: where are these results?

Response: Thank you. We apologize, no confocal microscope was used in the experiments presented in this article. Lines 154,166,167  – are revised (in red).

However, measuring of tophography of WC/C coating has been measured using atomic force microscopy … - Lines 154, 166, 167 – are revised (in red).

  • They prepared to type of samples: over Si and over steel, when are used each type?

Response: Thank you. Samples of monocrystal of Si have been used to evaluate the thickness and structure of WC/C coatings via scan electron microscopy (SEM). Samples made of steel have been used to evaluate the COF of WC/C coatings via Pin-on-disc test. Lines 92- 95 – are supplement and  revised (in red).

RESULTS AND DISCUSSION:

  • Line 213-215: it is said columns. However, they are not clear in such image. Neither AFM image provides data to assure such structure.

Response: Thank you. Lines 215-221 - are revised (in red).

  • Figure 9: the figure caption is not clear. X-ray diffraction scan? What is it?

Response: Thank you. We apologize,  Figure 9: the figure caption - Lines 353,354 - are revised (in red).

  • How has been the chemical analysis performed?

Response: Thank you. The chemical analysis has been performed using GDOES - Lines 157,158 - are revised (in red).

Reviewer 3 Report

Comments for Materials 868721-V1

Based on my experience in tribology. Some comments should be noted.

(1) In Fig. 4, the curves of real-time COF should be the type of point-line.

(2) In Fig. 8(b), the name of horizontal axis should be "annealing temperature" not only "temperature".

(3) The definition of error bars in Fig. 8(b) should be added in the manuscript.

(4) All the COF curves related in Fig. 8(b) should be presented in the manuscript.

Author Response

Response to Reviewer 3 Comments

Thank you very much for your kind evaluation of our work. We do agree with all your proposals and comments and have modified the manuscript according to them.

We hope that with your comments the manuscript will be suitable for publishing in Materials and will attract many potential readers of the journal.

Kind regards,

Authors.

Comments and Suggestions for Authors:

Comments for Materials 868721-V1

Based on my experience in tribology. Some comments should be noted.

(1) In Fig. 4, the curves of real-time COF should be the type of point-line.

Response: Thank you. Curves of real-time COF are the type of point-line. Fig. 4 (Line 256) is replaced by line 257 (in green)

(2) In Fig. 8(b), the name of horizontal axis should be "annealing temperature" not only "temperature".

Response: Thank you. In Fig. 8(b), the name of horizontal axis is revised (in green)

(3) The definition of error bars in Fig. 8(b) should be added in the manuscript.

Response: Thank you. The definition of error bars in Fig. 8(b) were added in the manuscript. Lines 315, 321-323, 325, 375, 380, 404 are revised (in green)

(4) All the COF curves related in Fig. 8(b) should be presented in the manuscript.

Response: Thank you. Due to clarity of the comparison of the measured values of COF we dare to suggest that Fig. 8(b) stays the same (without the presentation of all COF curves in the manuscript). Moreover, authors have evaluated only the achieved maximal values of COF, which were compared reciprocally, and those that were compared with results published in previous works (References 4 and 15), to which this work is referencing. In future, we will focuse our research on wear of evaluated coatings and the suggested can be implemented in these works.

Round 2

Reviewer 2 Report

Dear authors,

Thank you very much for the acclarations of the comments. There are still few questions to be addressed:

Line 92: it is scanning electron microscopy

Line 157-158: GDOES has been included but the methodology is not properly described.

Figure 9: are all of them SEM images?

Author Response

Kosice, 24-07-2020

Dear Editor,

please find uploaded a copy of the manuscript entitled "Microstructure and mechanical properties of annealed WC/C PECVD coatings deposited using hexacarbonyl of W with different gases" by P. Horňák, D. Kottfer, K. Kyziol, M. Trebuňová, J. Majerníková, L. Kaczmarek, J. Trebuňa, J. Hašuľ and M. Paľo.

We have revised the manuscript according to reviewer comments (Reviewer 2, Round 2) modifying certain parts of the work. As well, we introduced small changes in the text in order to improve the quality of the paper.

The revisions in the manuscript have been indicated in red color.

By this chance, we would like to thank you and the reviewers for their time and effort in improving our manuscript.

Sincerely,

Daniel Kottfer

Responses to Reviewer comments

Reviewer: Line 92: it is scanning electron microscopy

Answer: Authors agree with reviewer’s comment.

Author actions: Sentence (Lines 91-93) have been corrected as "Samples of a monocrystal of Si with approximate size 20x15x1 mm have been used to evaluate the thickness and structure of WC/C coatings via scanning electron microscopy (SEM).".

Reviewer: Line 157-158: GDOES has been included but the methodology is not properly described.

Answer: Authors agree with reviewer’s comment.

Author actions: Suggested changes have been made and the additional comments in section 2.4 (Lines 156-160) have been added: "Chemical analysis was evaluated as the depth of the chemical concentration profile using the Glow Discharge Optical Emission Spectroscopy (GDOES) technique. During this analysis, the Spectruma GDA 750 equipment has been applied,  which is provided with a monochromator with a Czerny-Turner design (focal length of 480 mm). The spatial resolution of the monochromator was 0.025 nm.".

Reviewer: Figure 9: are all of them SEM images?

Answer: Yes, all images presented in Fig. 9 were obtained using scanning electron microscopy (SEM).

Author actions: A caption of Fig. 9 have been corrected as : "Figure 9. Morphology of the surface and cross-sectional brake of WC/C coating (based on SEM images) deposited with N2 (added gas): a), b) RT and annealed at c), d) 500 °C, e), f) 800 °C".

Reviewer 3 Report

Thanks for your reply.

According to your response: Fig. 8(b) stays the same (without the presentation of all COF curves in the manuscript). Moreover, authors have evaluated only the achieved maximal values of COF.

However, based on your real-time COF curves in Fig. 4 (COF of WC/C coatings). The two curves are different, I cannot agree with you that Fig. 8(b) stays the same.

Hence, it is inappropriate to evaluate only the achieved maximal values of COF.

If possible, I hold the opinion that "All the COF curves related in Fig. 8(b) should be presented in the manuscript". All the curves can be illustrated and compared in one or two figures. At least, some typical COF curves should be presented. The friction properties could be then analyzed.

Author Response

Kosice, 03-08-2020

Dear Editor,

please find uploaded a copy of the manuscript entitled "Microstructure and mechanical properties of annealed WC/C PECVD coatings deposited using hexacarbonyl of W with different gases" by P. Horňák, D. Kottfer, K. Kyziol, M. Trebuňová, J. Majerníková, L. Kaczmarek, J. Trebuňa, J. Hašuľ, and M. Paľo.

We have revised the manuscript according to reviewer comments (Reviewer 3, Round 2) modifying certain parts of the work.

The revisions in the manuscript have been indicated in green color.

By this chance, we would like to thank you and the reviewers for their time and effort in improving our manuscript.

Sincerely,

Daniel Kottfer

Responses to Reviewer comments

Reviewer: According to your response: Fig. 8(b) stays the same (without the presentation of all COF curves in the manuscript). Moreover, authors have evaluated only the achieved maximal values of COF.

However, based on your real-time COF curves in Fig. 4 (COF of WC/C coatings). The two curves are different, I cannot agree with you that Fig. 8(b) stays the same.

Hence, it is inappropriate to evaluate only the achieved maximal values of COF.

If possible, I hold the opinion that "All the COF curves related in Fig. 8(b) should be presented in the manuscript". All the curves can be illustrated and compared in one or two figures. At least, some typical COF curves should be presented. The friction properties could be then analyzed.

Answer: Authors agree with the reviewer’s comment.

Author actions: Page 8, lines 259-261: Fig. 4 (with Figure caption) have been corrected.

Page 11, line 322: The obtained value of COF of the WC coating deposited with Ar and without annealing has been supplemented using the reference to Figure 4. COF value of the deposited WC/C coating with Ar has grown to 0.77 ± 0.02 (Figure 4), which …”

Pages 11-12, lines 332-336: Additional comments were added in section 3.2.2. of the manuscript: Ar has inhibited the degradation of WC coating by swelling (Figure 5b). That could have caused the creation of a smaller contact surface between WC coating and the counterpart (ball) during the Ball-on-disc test, which could have reduced the value of COF when compared with WC coating with N2 (Figure 8a).”

Page 12, lines 336-339: Fig. 8 has been supplemented by parts 8c, 8d, and additional information were added in captions of Fig. 8. Additional comments were added in section 3.2.2. of the manuscript: "A significant decrease of COF after annealing at the temperatures of 500 °C (see Fig. 8c, using N2) and 800 °C (see Fig. 8d, using Ar) could have been caused by the presence of C in the coating in the form of graphite."

Page 13, lines 346-348: This text has been corrected, the corrected part is this: "Figure 8. Dependence of HIT (a) and COF (b) vs. annealing temperature of the WC/C coatings, deposited using Ar or N2; COF curves vs. annealing temperature of the WC/C coatings deposited using N2 (c) or Ar (d)."
